# HYPER-PARAMETER TUNING FOR FAIR CLASSIFICATION WITHOUT SENSITIVE ATTRIBUTE ACCESS

## ABSTRACT

Fair machine learning methods seek to train models that balance model performance across demographic subgroups defined over sensitive attributes like race and gender. Although sensitive attributes are typically assumed to be known during training, they may not be available in practice due to privacy and other logistical concerns. Recent work has sought to train fair models without sensitive attributes on training data. However, these methods need extensive hyper-parameter tuning to achieve good results, and hence assume that sensitive attributes are known on validation data. However, this assumption too might not be practical. Here, we propose Antigone, a framework to train fair classifiers without access to sensitive attributes on either training or validation data. Instead, we generate pseudo sensitive attributes on the validation data by training a biased classifier and using the classifier's incorrectly (correctly) labeled examples as proxies for minority (majority) groups. Since fairness metrics like demographic parity, equal opportunity and subgroup accuracy can be estimated to within a proportionality constant even with noisy sensitive attribute information, we show theoretically and empirically that these proxy labels can be used to maximize fairness under average accuracy constraints. Key to our results is a principled approach to select the hyper-parameters of the biased classifier in a completely unsupervised fashion (meaning without access to ground truth sensitive attributes) that minimizes the gap between fairness estimated using noisy versus ground-truth sensitive labels.

## 1 INTRODUCTION

Deep neural networks have achieved state-of-the-art accuracy on many tasks including face recognition (Buolamwini & Gebru, 2018; Grother et al., 2010; Ngan & Grother, 2015), autonomous driving (Zhang et al., 2021; Chitta et al., 2021), medical image diagnosis (Litjens et al., 2017; Cheplygina et al., 2019), *etc*. But, prior work (Hovy & Søgaard, 2015; Oren et al., 2019; Hashimoto et al., 2018a) has found that state-of-the-art networks exhibit unintended biases towards specific population groups, especially harming minority groups. Seminal work by Buolamwini & Gebru (2018) demonstrated, for instance, that commercial face recognition systems had lower accuracy on darker skinned women than other groups. A body of work has sought to design fair machine learning algorithms that account for a model's performance on a per-group basis (Prost et al., 2019; Sagawa* et al., 2020; Liu et al., 2021; Sohoni et al., 2020).

Much of the prior work assume that demographic attributes like gender and race on which we seek to train a fair model, which we refer to as *sensitive attributes*, are available on training and validation data Sagawa* et al. (2020); Prost et al. (2019). However, there is a growing body of literature (Veale & Binns, 2017; Holstein et al., 2019) highlighting many real-world settings in which sensitive attributes may not be available. This is for multiple reasons. For example, the data subject may abstain from providing sensitive information to eschew potential discrimination in future (Markos et al., 2017). In other settings, the attributes on which the model discriminates might not even be known (Citron & Pasquale, 2014; Pasquale, 2015). For instance, in algorithmic hiring decisions, Köchling & Wehner (2020) highlight that bias and discrimination are recognized only after making real world decisions on applicants due to unknown attributes on which the model discriminates during training. Consequently, a large American e-commerce company had to cease using algorithmic tools for hiring purposes as it was unintentionally discriminating female applicants (Dastin, 2018).

Recent work seeks to train fair classifiers without access to sensitive attributes on the training set (Liu et al., 2021; Creager et al., 2021; Nam et al., 2020; Hashimoto et al., 2018a). The common theme across these methods is to up-weight misclassified examples either by splitting the training stage into two separate stages Liu et al. (2021) (identify mis-classified examples in stage 1 and upweight in stage 2) or by alternating between these stages across training epochs Nam et al. (2020) (identifying misclassified examples in one epoch and upweighting in the next). However, Liu et al. (2021) has shown that these methods are highly sensitive to choice of hyper-parameters; the up-weighting factor, for example, can have a large impact on the resulting model's fairness. Some methods, therefore, tune hyper-parameters assuming access to sensitive information on the validation dataset. In fact, without this information, Liu et al. (2021) observed that these methods sometimes do worse than using standard ERM. But, sensitive information on the validation dataset may not be available for the same reasons they are hard to acquire on training data.

In this paper, we propose Antigone, a simple, principled approach that enables hyper-parameter tuning for fairness without access to sensitive attributes on validation data. Antigone can be used to tune hyper-parameters for any prior method, for instance, JTT (Liu et al., 2021), LfF (Nam et al., 2020), CVaR DRO (Hashimoto et al., 2018a), that trains fair models without sensitive attributes on training data, and for several fairness metrics including demographic parity, equal opportunity and worst sub-group accuracy. We note that these prior methods also address the problem of spurious correlations (Sagawa et al., 2020; Wang & Culotta, 2021) and their impact on accuracy. As such, Antigone can also be used to address spurious correlations, but we focus on fairness in this paper.

Antigone builds on the same intuition as in prior work: mis-classified examples of a classifier trained with standard empirical risk serves as an effective proxy for minority groups. Accordingly, Antigone trains a *biased* classifier as a noisy sensitive attributes labeller on the validation data, labelling correctly and incorrectly classified examples as majority and minority groups, respectively. But this raises a key question: *how do we select the hyper-parameters of the noisy labeler?*

Intuitively, to maximize utility of the noisily labelled validation set, we seek to maximize the fraction of minority (majority) samples in the incorrect (correct) sets. Since this cannot be measured directly, Antigone instead maximizes the distance between the data distributions of the two sets, which we measure using the Euclidean distance between the means (EDM) of the two distributions. We provide theoretical justification for our choice under the mutually contaminated (MC) noise model (Scott et al., 2013) that assumes that a fraction of majority (minority) group labels are contaminated with labels from minority (majority) group. Lamy et al. (2019) et al. show that common fairness metrics can be estimated up to a proportionality constant under the MC model. We show that Antigone's EDM criteria maximizes this proportionality constant, thus providing the most reliable estimates of fairness.

We evaluate Antigone in conjunction with JTT Liu et al. (2021) on the CelebA, Waterbirds and Adult datasets which are commonly used in fairness literature. We compare Antigone with baselines that assume ground-truth knowledge of sensitive attributes and standard ERM on demographic parity, equal opportunity, and worst subgroup accuracy. Antigone significantly closes the fairness gap between standard ERM training and fairness with ground truth sensitive attributes. Compared with GEORGE that estimates majority/minority group labels by clustering the activations of an ERM model, Antigone produces more accurate labels and results in improved fairness. Ablation studies demonstrate the effectiveness of Antigone's EDM based hyper-parameter tuning.

## 2 PROPOSED METHODOLOGY

We now describe Antigone, starting with the problem formulation (Section 2.1) followed by a description of the Antigone algorithm (Section 2.2).

### 2.1 PROBLEM SETUP

Consider a data distribution over set $\mathcal{D} = \mathcal{X} \times \mathcal{A} \times \mathcal{Y}$, the product of input data ($\mathcal{X}$), sensitive attributes ($\mathcal{A}$) and target labels ($\mathcal{Y}$) triplets. We are given a training set $D^{tr} = \{x_i^{tr}, a_i^{tr}, y_i^{tr}\}_{i=1}^{N^{tr}}$ with $N^{tr}$ training samples, and a validation set $D^{val} = \{x_i^{val}, a_i^{val}, y_i^{val}\}_{i=1}^{N^{val}}$ with $N^{val}$ validation samples. We will assume binary sensitive attributes ($\mathcal{A} \in \{0, 1\}$) and target labels ($\mathcal{Y} \in \{0, 1\}$). We note that for now Antigone is limited to binary sensitive attributes, but can be extended to multiple target labels.

We seek to train a machine learning model, say a deep neural network (DNN), which can be represented as a parameterized function $f_\theta : \mathcal{X} \to \mathcal{Y} \in \{0, 1\}$, where $\theta \in \Theta$ are the trainable parameters, e.g., DNN weights and biases. Standard fairness unaware empirical risk minimization (ERM) optimizes over trainable parameters $\theta$ to minimize average loss $\mathcal{L}_{ERM}$:

$$\mathcal{L}_{ERM} = -\frac{1}{N^{tr}} \sum_{i=1}^{N} l(x_i^{tr}, y_i^{tr}), \tag{1}$$

on $D^{tr}$, where $l(x_i, y_i)$ is the binary cross-entropy loss.

Optimized model parameters $\theta^*$ are obtained by invoking a training algorithm, for instance stochastic gradient descent (SGD), on the training dataset and model, i.e., $\theta^{*,\gamma} = \mathcal{M}^{ERM}(D^{tr}, f_\theta, \gamma)$, where $\gamma \in \Gamma$ are hyper-parameters of the training algorithm including learning rate, training epochs etc. Hyper-parameters are tuned by evaluating models $f_{\theta*,\gamma}$ for all $\gamma \in \Gamma$ on $D^{val}$ and picking the best model. More sophisticated algorithms like Bayesian optimization can also be used.

Since standard ERM models suffer from unintended biases in their predictions, fair ML algorithms seek instead to optimize metrics that explicitly account for the performance on demographic sub-groups. We review three commonly used metrics below:

- **Demographic parity (DP):** DP requires the model's outcomes to be independent of sensitive attribute. In practice, we seek to minimize the demographic parity gap:
$$\Delta_\theta^{DP} = \mathbb{P}[f_{,}[f_\theta(X) = 1|A = 1] - \mathbb{P}[f_\theta(X) = 1|A = 0]). \tag{2}$$

  **Equal opportunity (EO):**   EO aims to equalize only the model's true positive rates across sensitive attributes. In practice, we seek to minimize
$$\Delta_\theta^{EO} = \mathbb{P}[f_\theta(X) = 1|A = 1, Y = 1] - \mathbb{P}[f_\theta(X) = 1|A = 0, Y = 1]. \tag{3}$$

- **Worst-group accuracy (WGA):** WGA seeks to maximize the minimum accuracy over all sub-groups over sensitive attributes and target labels. That is, we seek to maximize:
$$WGA_\theta = \min_{a \in \{0,1\}, y \in \{0,1\}} \mathbb{P}[f(x) = y|A = a, Y = y]. \tag{4}$$

In all three settings, we seek to train models that optimize fairness under a constraint on average accuracy. For example, for equal opportunity, we seek $\theta^* = \arg\min_{\theta \in \Theta} \Delta_\theta^{EO}$ such that $\mathbb{P}[f_\theta(x) = Y] \geq Acc$, where $Acc$ is a user specified constraint. With access to sensitive attributes, the fairness metric (and average accuracy) can be evaluated on the validation set. The challenge here is that sensitive attributes are unavailable.

## 2.2 ANTIGONE ALGORITHM

We now describe the Antigone algorithm which consists of three main steps. In step 1, we train multiple intentionally biased ERM models that each provide pseudo sensitive attribute labels on validation data. We view each model as a noisy sensitive attribute labeller on the validation set. In step 2, we use the proposed EDM metric to pick a noisy labeller from step 1 with the least noise. Finally, in step 3, we use the labelled validation set from step 2 to tune the hyper-parameters of methods like JTT that train fair classifiers without sensitive attributes on training data.

**Step 1: Generating sensitive attribute labels on validation set.**   In step 1, we use the training dataset and standard ERM training to obtain a set of classifiers, $\theta^{*,\gamma} = \mathcal{M}^{ERM}(D^{tr}, f_\theta, \gamma)$, each corresponding to a different value of training hyper-parameters $\gamma \in \Gamma$. As we discuss in Section 2.1, these include learning rate, weight decay and number of training epochs. Each classifier is used to generate pseudo sensitive attribute labels on the validation set by assigning correctly (incorrectly) classified examples to the majority (minority) groups. That is, each classifier yields a validation set

$$D^{val,\gamma} = \{x_i^{val}, a_i^{val,\gamma}, y_i^{val}\}_{i=1}^{N^{val}} \quad \forall \gamma \in \Gamma \tag{5}$$

where:

$$a_i^{val,\gamma} = \begin{cases} 1, & \text{if } f_{\theta*,\gamma}(x_i^{val}) = y_i^{val} \\ 0, & \text{otherwise.} \end{cases} \tag{6}$$

From these noisily labelled validation sets, we now seek to pick the one whose pseudo sensitive attribute labels match most closely with true (but unknown) sensitive attributes. That is, we seek to pick the hyper-parameters corresponding to the "best" noisy labeller.

**Step 2: Picking the best noisy labeller.** The noisy labellers in Step 1 partition inputs in the validation set into two sets containing correctly and incorrectly classified inputs. These serve as proxies for majority and minority groups, respectively. Specifically, let the correct set (or noisily labeled set of majority examples) be $X^{val,\gamma}_{A=1,noisy} = \{x^{val}_i : a^{val,\gamma}_i = 1\}$ and the incorrect set (or noisily labeled set of minority examples) be $X^{val,\gamma}_{A=0,noisy} = \{x^{val}_i : a^{val,\gamma}_i = 0\}$.

To estimate fairness accurately, we would like our noisy labeler to be biased, i.e., to place all majority (minority) group inputs in the correct (incorrect) set. In the absence of true sensitive attribute labels, we can measure bias using the distance between the data distributions in the correct and incorrect sets. In Antigone, we pick the simplest distance metric between two distributions, i.e., the Euclidean distance between their means (EDM). Formally,

$$EDM^\gamma = \|\mu(X^{val,\gamma}_{A=1,noisy}) - \mu(X^{val,\gamma}_{A=0,noisy})\|_2 \tag{7}$$

where $\mu(.)$ represents the empirical mean of a dataset. In Section 2.3 we theoretically justify this choice. We pick $\gamma^* = \arg\max_{\gamma \in \Gamma} EDM^\gamma$. Note that in practice we pick two different noisy labellers corresponding to target labels $Y = \{0, 1\}$.

**Step 3: Training a fair model.** Step 2 yields $D^{val,\gamma^*}$, a validation dataset with (estimated) sensitive attribute labels. We can provide $D^{val,\gamma^*}$ as an input to any method that trains fair models without access to sensitive attributes on training data, but requires a validation set with sensitive attribute labels to tune its own hyper-parameters. In our experimental results, we use $D^{val,\gamma^*}$ to tune the hyper-parameters of JTT (Liu et al., 2021) and GEORGE (Sohoni et al., 2020). We note that GEORGE proposes its own method to obtain sensitive attributes labels on validation data, but replacing it with Antigone improves on GEORGE's performance.

## 2.3 Analyzing Antigone under MC Noise

Prior work Lamy et al. (2019) has modeled noisy sensitive attributes using the mutually contaminated (MC) noise model Scott et al. (2013). Here, it is assumed that we have access to noisy datasets, $D_{A=0,noisy}$ and $D_{A=1,noisy}$, corresponding to minority and majority groups, respectively, that are mixtures of their ground-truth datasets $D_0$ and $D_1$. Specifically,

$$D_{A=1,noisy} = (1-\alpha)D_{A=1} + \alpha D_{A=0}$$

$$D_{A=0,noisy} = \beta D_{A=1} + (1-\beta)D_{A=0} \tag{8}$$

where $\alpha$ and $\beta$ are noise parameters. Note that strictly speaking Equation 8 should refer to the probability distributions of the respective datasets, but we will abuse this notation to refer to the datasets themselves. As such Equation 8 says that fraction $\alpha$ of the noisy majority group, $D_{A=1,noisy}$, is contaminated with data from the minority group, and fraction $\beta$ of the noisy minority group, $D_{A=0,noisy}$, is contaminated with data from the majority group. An extension of this model assumes that the noise parameters are target label dependent, i,e., $(\alpha_0,\beta_0)$ for $Y = 0$ and $(\alpha_1,\beta_1)$ for $Y = 1$.

Note that the MC model assumes that noisy datasets are constructed by sampling independently from the ground-truth distributions. While this is not strictly true in our case since the noise in our sensitive attribute labels might be instance dependent, the MC model can still shed light on the design of Antigone.

**Proposition 1.** *(Lamy et al., 2019) Under the MC noise model in Equation 8, demographic parity and equal opportunity gaps measured on the noisy datsets are proportional to the true DP and EO gaps. Mathematically:*

$$\Delta^{DP}(D_{A=0,noisy} \cup D_{A=1,noisy}) = (1-\alpha-\beta)\Delta^{DP}(D_{A=0} \cup D_{A=1}), \tag{9}$$

*and*

$$\Delta^{EO}(D_{A=0,noisy} \cup D_{A=1,noisy}) = (1-\alpha_1-\beta_1)\Delta^{EO}(D_{A=0} \cup D_{A=1}). \tag{10}$$

From Equation 9 and Equation 10, we can conclude that under the MC noise model, the DP and EO gaps can be equivalently minimized using noisy sensitive attribute labels, assuming independent

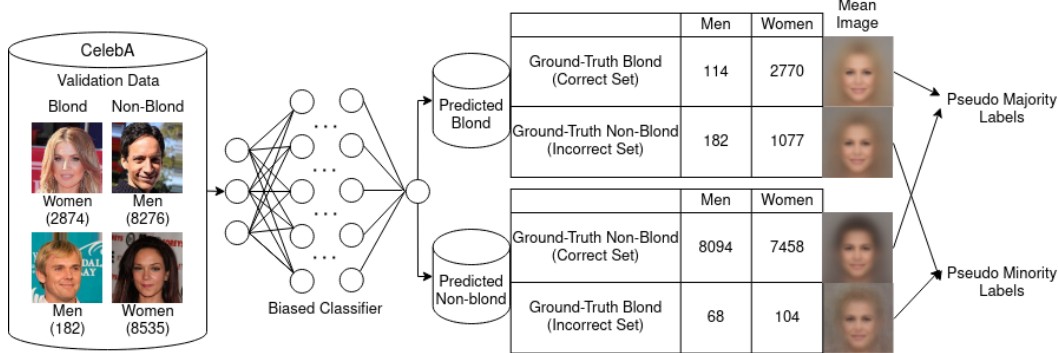

Figure 1: Figure illustrates the schematic of Antigone generating pseudo sensitive attributes on CelebA validation data by training a biased classifier and using the classifier's incorrectly (correctly) labeled examples as proxies for minority (majority) groups.

contamination and infinite validation data samples. In practice, these assumptions do not hold, however, and therefore we seek to maximize the proportionality constant $1 - \alpha - \beta$ (or $1 - \alpha_1 - \beta_1$) to minimize the gap between the true and estimated fairness values.

**Lemma 1.** *Assume $X_{A=0,noisy}$ and $X_{A=1,noisy}$ correspond to the input data of noisy datasets in the MC model. Then, maximizing the EDM between the $X_{A=0,noisy}$ and $X_{A=1,noisy}$, i.e., $\|\mu(X_{A=0,noisy}) - \mu(X_{A=1,noisy})\|_2$ maximizes $1 - \alpha - \beta$.*

*Proof.* From Equation 8, we can see that $\|\mu(X_{A=0,noisy}) - \mu(X_{A=1,noisy})\|_2 = (1 - \alpha - \beta)^2 \|\mu(X_{A=0}) - \mu(X_{A=1})\|_2$. Here $\|\mu(X_{A=0}) - \mu(X_{A=1})\|_2$ is the EDM between the ground truth majority and minority data and is therefore a constant. Hence, maximizing EDM between $X_{A=0,noisy}$ and $X_{A=1,noisy}$ maximizes $1 - \alpha - \beta$. $\qquad\square$

In practice, we separately maximize EDM for target labels $Y = \{0, 1\}$ and hence maximize both $1 - \alpha_0 - \beta_0$ and $1 - \alpha_1 - \beta_1$. We note that our theoretical justification motivates the use of EDM for DP and EO fairness. While not exact, minimizing $\alpha + \beta$ using EDM as a proxy is still helpful for WGA because it reduces contamination and, empirically, provides more reliable estimates for sub-group accuracy.

## 3 EXPERIMENTAL SETUP

We empirically evaluate Antigone on the CelebA, Waterbirds and UCI Adult datasets which have been extensively studied in fairness literature. In this section, we present the details about Antigone's implementation, evaluation and network architecture used for these three datasets. To begin, we note Antigone can be deployed in conjunction with any method that trains fair classifiers without sensitive attributes on training data. We evaluate Antigone with one such state-of-the-art method, JTT (Liu et al., 2021). We begin by briefly describing how Antigone is deployed in conjunction with JTT.

**JTT+Antigone:** JTT operates in two stages. In the first stage, a biased model is trained using $T$ epochs of standard ERM training to identify the incorrectly classified training examples. In the second stage, the misclassified examples are upsampled $\lambda$ times and the model is trained again to completion with standard ERM. The hyperparameters of stage 1 and stage 2 classifiers, including early stopping epoch $T$ and upsampling factor $\lambda$, are jointly tuned using a validation dataset with ground-truth sensitive attribute labels. We replace the ground-truth validation dataset with noisy sensitive attributes obtained from Antigone.

### 3.1 CELEBA DATASET

**Dataset details:** CelebA (Liu et al., 2015) is an image dataset, consisting of 202,599 celebrity face images annotated with 40 attributes including gender, hair colour, age, smiling, etc. The task is to predict hair color, which is either blond $Y = 1$ or non-blond $Y = 0$ and the sensitive attribute is

gender $A = \{\text{Men}, \text{Women}\}$. The dataset is split into training, validation and test sets with 162770, 19867 and 19962 images, respectively. Only 15% of individuals in the dataset are blond, and only 6% of blond individuals are men. Consequently, the baseline ERM model under-performs on the blond men.

**Hyper-parameter settings:** In all our experiments using CelebA dataset, we fine-tune a pre-trained ResNet50 architecture for a total of 50 epochs using SGD optimizer and a batch size of 128. We tune Antigone and JTT over three pairs of learning rates and weight decays, $(1e-04, 1e-04), (1e-04, 1e-02), (1e-05, 1e-01)$, which are also the values used in JTT. For Antigone, we also explore early stopping at any of the 50 training epochs. Antigone's hyper-parameters are tuned using the EDM approach. For JTT, we explore over $T \in \{1, 2, 5, 10, 15, 20, 25, 30, 35, 40, 45, 50\}$ and $\lambda \in \{20, 50, 100\}$ as reported in their paper. JTT's hyper-parameters are tuned using the validation dataset produced by Antigone. We report results for DP, EO and WGA fairness metrics. In each case, we seek to optimize fairness while constraining average accuracy to ranges $\{[90, 91), [91, 92), [92, 93), [93, 94), [94, 95)\}$.

### 3.2 WATERBIRDS DATASET

**Dataset details:** Waterbirds is a synthetically generated dataset, containing 11,788 images of water and land birds overlaid on top of either water or land backgrounds (Sagawa* et al., 2020). The task is to predict the bird type, which is either a waterbird $Y = 1$ or a landbird $Y = 0$ and the sensitive attribute is the background $A = \{\text{Water background}, \text{Land background}\}$. The dataset is split into training, validation and test sets with 4795, 1199 and 5794 images, respectively. While the validation and test sets are balanced within each target class, the training set contains a majority of waterbirds (landbirds) in water (land) backgrounds and a minority of waterbirds (landbirds) on land (water) backgrounds. Thus, the baseline ERM model under-performs on the minority group.

**Hyper-parameter settings:** In all our experiments using Waterbirds dataset, we train a pre-trained ResNet50 architecture for a total of 300 epoch using the SGD optimizer and a batch size of 64. We tune Antigone and JTT over three pairs of learning rates and weight decays, $(1e-03, 1e-04), (1e-04, 1e-01), (1e-05, 1.0)$, which are also the values used in JTT. For JTT, we explore over $T \in \{25, 40, 50, 60, 75, 100, 125, 150, 175, 200, 225, 250, 275, 300\}$ and $\lambda \in \{20, 50, 100\}$ as reported in their paper. In each case, we seek to optimize fairness while constraining average accuracy to ranges $\{[94, 94.5), [94.5, 95), [95, 95.5), [95.5, 96), [96, 96.5)\}$.

### 3.3 BASELINES FOR COMPARISONS

We evaluate JTT+Antigone against several baselines.

**Standard ERM:** A naive baseline is a standard ERM model that only seeks to maximize average accuracy and does not consider fairness. We train standard ERM models using the same network architectures and training hyper-parameters used for Antigone + JTT as reported above.

**JTT + Ground-truth sensitive attributes:** An upper bound for JTT+Antigone is a JTT model trained with ground-truth sensitive attributes on validation data, i.e., the overall approach used in JTT. For this, we used the reference implementations provided by JTT.

**GEORGE Sohoni et al. (2020):** GEORGE is a competing approach to Antigone that does not assume access to sensitive attributes on either training or validation data. GEORGE operates in two stages: In stage 1, an ERM model is trained until completion on the ground-truth target labels. The activation in the penultimate layer of the ERM model are clustered into $k$ clusters to generate pseudo sensitive attributes on both the training and validation datasets. These pseudo sensitive attributes are used to train and tune the hyper-parameters of a Group DRO model Sagawa* et al. (2020).

For a fair comparison with GEORGE, we replace its stage 1 with Antigone, and use the resulting pseudo sensitive attribute labels to tune the hyper-parameters of a Group DRO model Sagawa* et al. (2020). We refer to this approach as GEORGE+Antigone.

## 4 EXPERIMENTAL RESULTS

We now discuss the results of our empirical evaluations. We begin by analyzing the quality of sensitive attribute labels produced by Antigone and evaluate JTT/GEORGE+Antigone.

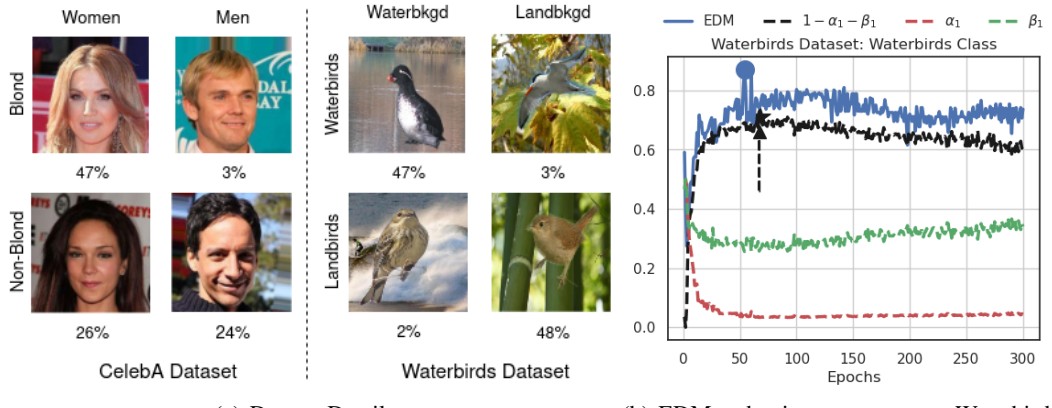

(a) Dataset Details     (b) EDM and noise parameters on Waterbirds

Figure 2: Figure (a) illustrates CelebA and Waterbirds datasets along with fraction of each sub-group examples in their respective training dataset. Figure (b) shows Euclidean Distance between Means (EDM) and noise parameters ($\alpha_1, \beta_1$ and $1 - \alpha_1 - \beta_1$) for the positive target class of Waterbirds dataset. The noise parameters are unknown in practice. Blue dot indicates the model that we pick to generate pseudo sensitive attributes, while black star indicates the model that maximizes $1 - \alpha_1 - \beta_1$.

**Quality of Antigone's sensitive attribute labels:** Antigone seeks to generate accurate sensitive attribute labels on validation data based on the EDM criterion (Lemma 1). In Figure 2(b), we empirically validate Lemma 1 by plotting EDM and noise parameters $\alpha_1$ (contamination in minority group labels), $\beta_1$ (contamination in minority group labels) and $1 - \alpha_1 - \beta_1$ (proportionality constant between true and estimated fairness) on Waterbirds dataset (similar plot for CelebA dataset is in Appendix Figure 3). As per Lemma 1, EDM allows us to maximize $1 - \alpha_1 - \beta_1$ since it is ideally proportional to this value. From the figure, we observe that in both cases the EDM metric indeed captures the trend in $1 - \alpha_1 - \beta_1$, enabling early stopping at an epoch that minimizes contamination.

The early stopping points based on EDM and oracular knowledge of $1 - \alpha_1 - \beta_1$ are shown in a blue dot and star, respectively. For Waterbirds these are very close.

Next, we evaluate the F1 scores of Antigone's noisy sensitive attributes for all four subgroups in the CelebA and Water-

Table 1: F1 Scores and Accuracy of the pseudo sensitive attribute labels generated by Antigone (w/o EDM), GEORGE, GEORGE with $k = 2$ clusters and Antigone (w/ EDM) on CelebA and Waterbirds datasets. Bold indicates higher F1 Scores/Accuracy across different methods. For CelebA, BM, BW, NBW and NBM refer to Blond Men, Blond Women, Non-blond Women and Non-blond Men, respectively. For Waterbirds, WL, WW, LW and LL refer to waterbirds landbkgd, waterbirds waterbkgd, landbirds waterbkgd and landbirds landbkgd, respectively.

| | Antigone (w/o EDM) | GEORGE | GEORGE ($k = 2$) | Antigone (w/ EDM) |
|---|---|---|---|---|
| | | CelebA (F1 Scores) | | |
| BM | $0.28 \pm 0.01$ | $0.13 \pm 0.02$ | $0.12 \pm 0.01$ | $\mathbf{0.35 \pm 0.04}$ |
| BW | $0.95 \pm 0.01$ | $0.43 \pm 0.04$ | $0.51 \pm 0.02$ | $\mathbf{0.96 \pm 0.00}$ |
| NBW | $0.22 \pm 0.02$ | $0.42 \pm 0.01$ | $\mathbf{0.6 \pm 0.01}$ | $0.22 \pm 0.01$ |
| NBM | $0.67 \pm 0.01$ | $0.4 \pm 0.02$ | $0.31 \pm 0.01$ | $\mathbf{0.68 \pm 0.01}$ |
| Acc. | $0.59 \pm 0.01$ | $0.33 \pm 0.01$ | $0.48 \pm 0.00$ | $\mathbf{0.60 \pm 0.00}$ |
| | | Waterbirds (F1 Scores) | | |
| WL | $0.41 \pm 0.02$ | $0.43 \pm 0.02$ | $0.52 \pm 0.01$ | $\mathbf{0.76 \pm 0.03}$ |
| WW | $0.72 \pm 0.00$ | $0.36 \pm 0.02$ | $0.43 \pm 0.02$ | $\mathbf{0.83 \pm 0.01}$ |
| LW | $0.58 \pm 0.02$ | $0.44 \pm 0.03$ | $0.55 \pm 0.03$ | $\mathbf{0.78 \pm 0.04}$ |
| LL | $0.76 \pm 0.01$ | $0.34 \pm 0.02$ | $0.55 \pm 0.03$ | $\mathbf{0.84 \pm 0.02}$ |
| Acc. | $0.68 \pm 0.01$ | $0.30 \pm 0.02$ | $0.53 \pm 0.03$ | $\mathbf{0.81 \pm 0.02}$ |

birds datasets. In Table 1 we compare Antigone's F1 Score to GEORGE with the baseline number of clusters and GEORGE with $k = 2$ clusters. Across CelebA and Waterbirds datasets and all four subgroups, we find that Antigone outperforms GEORGE except for one sub-group in CelebA dataset. In the Appendix Table 4, we also include precision and recall values and reach similar conclusion. Additionally, we note that the pseudo label generated by Antigone are 1.25 and 1.5 times more accurate than GEORGE's pseudo labels on CelebA and Waterbirds datasets, respectively.

To understand the benefits of the proposed EDM metric, we implement a version of Antigone but tune hyper-parameters using standard ERM. We refer to this as Antigone (w/o EDM) and find in Table 1 that Antigone with EDM outperforms the version without EDM. We later report on the fairness achieved by these different versions.

Table 2: (Avg. Accuracy, Fairness) on test data for different validation accuracy thresholds on the CelebA dataset. Lower DP and EO gaps are better. Higher WGA is better.

| Val. Thresh. | Method | DP Gap | EO Gap | Worst-group Acc. |
|---|---|---|---|---|
| [94, 95) | Antigone + JTT | $(94.6, 15.0) \pm (0.2, 0.7)$ | $(94.7, 30.1) \pm (0.2, 3.2)$ | $(94.4, 59) \pm (0.2, 4.7)$ |
| | Ground-Truth + JTT | $(94.7, 14.9) \pm (0.2, 0.6)$ | $(94.5, 30.4) \pm (0.2, 2.3)$ | $(94.3, 62.1) \pm (0.3, 3.2)$ |
| [93, 94) | Antigone + JTT | $(93.7, 13.1) \pm (0.2, 0.7)$ | $(93.6, 26.4) \pm (0.4, 5.0)$ | $(93.4, 62.6) \pm (0.2, 7.0)$ |
| | Ground-Truth + JTT | $(93.6, 13.1) \pm (0.1, 0.6)$ | $(93.6, 22.7) \pm (0.3, 2.7)$ | $(93.4, 67.9) \pm (0.1, 1.9)$ |
| [92, 93) | Antigone + JTT | $(92.7, 11.1) \pm (0.2, 0.5)$ | $(92.3, 20.2) \pm (0.2, 3.4)$ | $(92.7, 68.1) \pm (0.4, 3.7)$ |
| | Ground-Truth + JTT | $(92.7, 11.2) \pm (0.3, 0.5)$ | $(92.7, 16.9) \pm (0.4, 2.9)$ | $(92.7, 72.5) \pm (0.2, 1.3)$ |
| [91, 92) | Antigone + JTT | $(91.7, 9.6) \pm (0.1, 0.5)$ | $(91.5, 16.3) \pm (0.3, 3.4)$ | $(91.3, 63.2) \pm (0.3, 2.6)$ |
| | Ground-Truth + JTT | $(91.8, 9.7) \pm (0.2, 0.5)$ | $(91.8, 10.1) \pm (0.3, 4.1)$ | $(91.8, 77.3) \pm (0.1, 2.4)$ |
| [90, 91) | Antigone + JTT | $(91.0, 8.3) \pm (0.2, 0.4)$ | $(90.9, 13.1) \pm (0.1, 3.6)$ | $(90.9, 63.1) \pm (0.5, 4.4)$ |
| | Ground-Truth + JTT | $(91.0, 8.4) \pm (0.2, 0.4)$ | $(90.7, 6.8) \pm (0.4, 3.7)$ | $(91.4, 78.6) \pm (0.2, 2.0)$ |
| | ERM | $(95.8, 18.6) \pm (0.0, 0.3)$ | $(95.8, 46.4) \pm (0.0, 2.2)$ | $(95.8, 38.7) \pm (0.0, 2.8)$ |

**Antigone + JTT:** Next, in Table 2, compare the test accuracy and fairness achieved by Antigone with JTT (Antigone + JTT) versus a baseline ERM model and with JTT using ground-truth sensitive attributes (Ground-Truth+JTT). As expected, baseline ERM yields unfair outcomes on all three fairness metrics: DP, EO and WGA. We observe that Antigone + JTT improves fairness over the baseline ERM model and closes the gap with Ground-Truth+JTT.

On DP, Antigone + JTT is very close to Ground-Truth + JTT results. Antigone + JTT substantially improve upon the EO Gap achieved by standard ERM. Antigone + JTT improves WGA from 38.7% using standard ERM to 68.1% at the expense of 3% accuracy drop. Ground-Truth + JTT improves WGA further up to 78.6% but with a larger average accuracy drop. Antigone + JTT achieves highest fairness for relatively high average accuracy values, although one would expect fairness to reduce with higher average accuracy. We believe this in part due to the noise in sensitive attribute labels that Antigone generates. Data for Adults (shown in the Appendix Table 8) and Waterbirds (shown in the Appendix Table 6) have the same trends.

**Comparison with GEORGE:** Like Antigone, GEORGE also generates pseudo-sensitive attributes on validation data, but as we noted in Table 1, Antigone's labels are more accurate and have higher F1 scores for different sub-groups. We now compare the fairness in terms of WGA achieved by GEORGE versus Antigone +GEORGE in which we

Table 3: Performance of GEORGE using Antigone's noisy validation data compared with GEORGE by itself. We observe that on CelebA and Waterbirds dataset, GEORGE + Antigone out-performs GEORGE, even if GEORGE assumes knowledge of number of clusters ($k = 2$) in its clustering step. We show errors for over five runs.

| Method | CelebA | | Waterbirds | |
|---|---|---|---|---|
| | Avg Acc | WGA | Avg Acc | WGA |
| ERM | $95.7 \pm 0.1$ | $34.5 \pm 3.1$ | $95.9 \pm 0.2$ | $29.7 \pm 1.6$ |
| GEORGE | $93.6 \pm 0.3$ | $60.4 \pm 2.3$ | $95.5 \pm 0.7$ | $50.0 \pm 5.8$ |
| Antigone + GEORGE | $93.3 \pm 0.3$ | $62.1 \pm 1.2$ | $96.0 \pm 0.2$ | $57.4 \pm 6.6$ |
| GEORGE (k=2) | $94.6 \pm 0.1$ | $62.6 \pm 2.1$ | $95.0 \pm 0.8$ | $46.7 \pm 11.7$ |
| Antigone + GEORGE (k=2) | $94.2 \pm 0.3$ | $65.3 \pm 2.9$ | $95.8 \pm 0.6$ | $54.4 \pm 7.1$ |

use Antigone's labeled validation data to tune GEORGE's training algorithm. Antigone + GEORGE is more fair than GEORGE alone on CelebA and Waterbirds (Table 3). On CelebA, Antigone + GEORGE has a small drop in average accuracy compared to GEORGE, while on Waterbirds, both average accuracy and fairness are better. In interpreting the errors on Waterbirds dataset, we should note that Antigone + GEORGE's WGA is atleast 8.4% higher than GEORGE's in each one of our multiple runs, except in one run where GEORGE has 1% higher WGA compared to Antigone + GEORGE, whereas on CelebA, Antigone + GEORGE's WGA was equal to or better than GEORGE's in each one of our multiple runs. Similar trend holds for Antigone + GEORGE ($k = 2$) and GEORGE ($k = 2$).

**Impact of EDM metric on fairness:** We already noted in Table 1 Antigone with the proposed EDM metric produces higher quality sensitive attribute labels compared to a version of Antigone that picks hyper-parameters using standard ERM. We evaluated these two approaches using JTT's training algorithm and find that Antigone with EDM results in a 5.7% increase in WGA and a small 0.06% increase in average accuracy.

## 5 RELATED WORKS

Several works have observed that standard ERM training algorithms can achieve state-of-the-art accuracy on many tasks, but unintentionally make biased predictions for different sensitive attributes failing to meet the fairness objectives (Hovy & Søgaard, 2015; Oren et al., 2019; Hashimoto et al., 2018a; Buolamwini & Gebru, 2018).

Fairness objectives can be broadly categorized into two types: individual fairness and group fairness. Individual fairness (Dwork et al., 2012; Kusner et al., 2017) requires similar individual to be treated similarly. Whereas, group fairness Prost et al. (2019); Quadrianto et al. (2019); Hardt et al. (2016) requires the groups of individuals divided based on a sensitive attribute like race, gender, etc., be treated equally. In this paper, we focus on the popular group fairness notions that include Demographic Parity, Equal Opportunity and Worst-group performance.

Methods that seek to achieve group fairness are three types: pre-processing, in-processing and post-processing algorithms. Pre-processing (Quadrianto et al., 2019; Ryu et al., 2018) methods focus on curating the dataset that includes removal of sensitive information or balancing the datasets. In-processing methods (Hashimoto et al., 2018b; Agarwal et al., 2018; Zafar et al., 2019; Lahoti et al., 2020; Prost et al., 2019; Liu et al., 2021; Sohoni et al., 2020) alter the training mechanism by using adding fairness constrains to the loss function or by training an adversarial framework to make predictions independent of sensitive attributes Zhang et al. (2018). Post-processing methods (Hardt et al., 2016; Wang et al., 2020; Savani et al., 2020) alter the outputs, for *e.g.* use different threshold for different sensitive attributes. In this work, we focus on in-processing algorithms.

Prior in-processing algorithms, including the ones referenced above, assume access to sensitive attributes on the training data and validation dataset. Recent work sought to train fair model without training data annotations Liu et al. (2021); Nam et al. (2020); Hashimoto et al. (2018a); Creager et al. (2021) but, except for GEORGE Sohoni et al. (2020), require sensitive attributes on validation dataset to tune the hyperparameters. Like GEORGE, we seek to train fair classification models without ground-truth sensitive information on either training or validation dataset.

Antigone is different from GEORGE in three ways: (1) Unlike GEORGE, we account for the model prediction and the ground-truth target label to generate pseudo-sensitive attributes. (2) The hyperparameters of the clustering step in GEORGE are fixed from literature and not specifically tuned for each dataset. In this paper, we propose a more principled approach to tune the model's hyperparameters in an unsupervised fashion to obtain noisy sensitive features. And finally, (3) GEORGE only focuses on worst-group accuracy, whereas Antigone can be adapted to different notions of fairness.

A related body of work develops *post-processing* methods to improve fairness without access to sensitive attributes but assuming a small set of labelled data for auditing Kim et al. (2019). One could use Antigone to create this auditing dataset, albeit with noise. Evaluating Antigone with these post-processing methods is an important avenue for future work.

## 6 CONCLUSION

In this paper, we propose Antigone, a method to enable hyper-parameter tuning for fair ML models without access to sensitive attributes on training or validation sets. Antigone generates high-quality pseudo-sensitive attribute labels on validation data by training a family of biased classifiers using standard ERM and using correctly (incorrectly) classified examples as proxies for majority (minority) group membership. We propose a novel EDM metric based approach to pick the most biased model from this family and provide theoretical justification for this choice using the MC noise model. The resulting validation dataset with pseudo sensitive attribute labels can then be used to tune the hyper-parameters of a fair training algorithm like JTT or GEORGE. We show that Antigone produces the highest precision sensitive attributes compared to the state-of-art. Future work will also seek to address the variance in fairness metrics (Mozannar et al., 2020) introduced by finite sample size under the MC noise model, extend Antigone to multiple-sensitive attributes, inter-sectional fairness and active learning of sensitive attributes.

## AVAILABILITY

Code with README.txt file is available at: `https://anonymous.4open.science/r/fairness_without_demographics-3BD0/README.md`

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

# A    APPENDIX

## A.1    UCI ADULT DATASET

**Dataset details:** Adult dataset (Dua & Graff, 2017) is used to predict if an individual's annual income is $\leq 50K$ ($Y = 0$) or $> 50K$ ($Y = 1$) based on several continuous and categorical attributes like the individual's education level, age, gender, occupation, etc. The sensitive attribute is gender $A = \{\text{Men}, \text{Women}\}$ Zemel et al. (2013). The dataset consists of 45,000 instances and is split into training, validation and test sets with 21112, 9049 and 15060 instances, respectively.

**Hyper-parameter settings:** In all our experiments using Adult dataset, we train a multi-layer neural network with one hidden layer consisting of 64 neurons. We train for a total of 100 epochs using the SGD optimizer and a batch size of 256. We tune Antigone and JTT by performing grid search over learning rates $\in \{1e-03, 1e-04, 1e-05\}$ and weight decays $\in \{1e-01, 1e-03\}$. For JTT, we explore over $T \in \{1, 2, 5, 10, 15, 20, 30, 35, 40, 45, 50, 65, 80, 95\}$ and $\lambda \in \{5, 10, 20\}$. In each case, we seek to optimize fairness while constraining average accuracy to ranges $\{[82, 82.5], [81.5, 82), [81, 81.5), [80.5, 81), [80, 80.5)\}$.

## A.2    QUALITY OF ANTIGONE'S SENSITIVE ATTRIBUTE LABELS

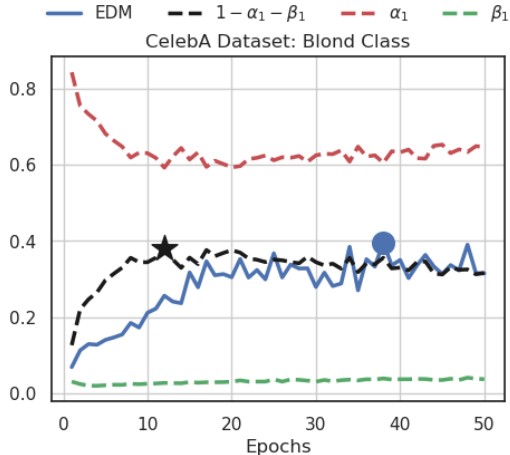

Figure 3: Euclidean Distance between Means (EDM) and noise parameters $\alpha_1, \beta_1$ and and $1 - \alpha_1 - \beta_1$ for the positive target class of CelebA dataset. The noise parameters are unknown in practice. Blue dot indicates the model that we pick to generate pseudo sensitive attributes, while black star indicates the model that maximizes $1 - \alpha_1 - \beta_1$.

Table 4: We tabulate the precision, recall, F1-score of the noisy validation groups generated from ERM model, GEORGE, GEORGE with number of clusters = 2 and Antigone. We observe that Antigone has higher precision and F1 scores across different noisy groups on CelebA and Waterbirds, respectively.

| | Antigone (w/o EDM) | GEORGE | GEORGE ($k = 2$) | Antigone (w/ EDM) |
|---|---|---|---|---|
| | CelebA (Precision, Recall, F1 Scores) | | | |
| Blond Men | 0.26, 0.31, 0.28 (0.02, 0.03, 0.01) | 0.09, 0.32, 0.13 (0.01, 0.09, 0.02) | 0.06, 0.70, 0.12 (0.01, 0.05, 0.01) | 0.36, 0.34, 0.35 (0.05, 0.04, 0.04) |
| Blond Women | 0.95, 0.94, 0.95 (0.01, 0.01, 0.01) | 0.94, 0.28, 0.43 (0.01, 0.04, 0.04) | 0.95, 0.35, 0.51 (0.00, 0.02, 0.02) | 0.96, 0.96, 0.96 (0.0, 0.0, 0.0) |
| Non-blond Women | 0.82, 0.13, 0.22 (0.01, 0.01, 0.02) | 0.51, 0.36, 0.42 (0.00, 0.01, 0.01) | 0.5, 0.76, 0.6 (0.00, 0.01, 0.01) | 0.86, 0.13, 0.22 (0.01, 0.01, 0.01) |
| Non-blond Men | 0.52, 0.97, 0.67 (0.00, 0.00, 0.01) | 0.53, 0.33, 0.40 (0.01, 0.02, 0.02) | 0.47, 0.23, 0.31 (0.01, 0.01, 0.01) | 0.52, 0.98, 0.68 (0.0, 0.0, 0.0) |
| CelebA Accuracy | $0.59 \pm 0.01$ | $0.33 \pm 0.01$ | $0.48 \pm 0.00$ | $0.60 \pm 0.00$ |
| | Waterbirds (Precision, Recall, F1 Scores) | | | |
| Waterbirds Landbkgd | 0.94, 0.26, 0.41 (0.01, 0.02, 0.02) | 0.56, 0.34, 0.43 (0.03, 0.02, 0.02) | 0.48, 0.57, 0.52 (0.01, 0.01, 0.01) | 0.96, 0.63, 0.76 (0.01, 0.04, 0.03) |
| Waterbirds Waterbkgd | 0.57, 0.98, 0.72 (0.01, 0.00, 0.00) | 0.55, 0.27, 0.36 (0.07, 0.03, 0.02) | 0.48, 0.39, 0.43 (0.02, 0.02, 0.02) | 0.73, 0.97, 0.83 (0.02, 0.01, 0.01) |
| Landbirds Waterbkgd | 0.96, 0.42, 0.58 (0.00, 0.03, 0.02) | 0.57, 0.36, 0.44 (0.04, 0.03, 0.03) | 0.55, 0.55, 0.55 (0.03, 0.03, 0.03) | 0.97, 0.65, 0.78 (0.00, 0.05, 0.04) |
| Landbirds Landbkgd | 0.63, 0.98, 0.76 (0.01, 0.00, 0.01) | 0.55, 0.24, 0.34 (0.04, 0.04, 0.02) | 0.55, 0.56, 0.55 (0.03, 0.04, 0.03) | 0.74, 0.98, 0.84 (0.03, 0.00, 0.02) |
| Waterbirds Accuracy | $0.68 \pm 0.01$ | $0.30 \pm 0.02$ | $0.53 \pm 0.03$ | $0.81 \pm 0.02$ |

Table 5: We tabulate the precision, recall, F1-score, accuracy of the noisy validation groups generated by varying the fraction of minority group examples in each class of CelebA dataset. We observe that Antigone has higher precision, recall, F1 score and accuracy if the imbalance is more in the training dataset.

| Fraction Minority | 5% | 20% | 35% | 50% |
|---|---|---|---|---|
| | Precision, Recall, F1 Score | | | |
| Blond Men (Minority) | 0.57, 0.40, 0.47 | 0.81, 0.17, 0.29 | 0.67, 0.15, 0.24 | 0.75, 0.14, 0.24 |
| Blond Women (Majority) | 0.97, 0.98, 0.98 | 0.83, 0.99, 0.90 | 0.68, 0.96, 0.79 | 0.53, 0.95, 0.68 |
| $1 - \alpha_1 - \beta_1$ | 0.54 | 0.64 | 0.35 | 0.28 |
| Blond Acc. | 0.95 | 0.83 | 0.68 | 0.55 |
| Non-blond Women (Minority) | 0.45, 0.26, 0.33 | 0.59, 0.17, 0.26 | 0.63, 0.18, 0.28 | 0.63, 0.16, 0.26 |
| Non-blond Men (Majority) | 0.96, 0.98, 0.97 | 0.82, 0.97, 0.89 | 0.68, 0.94, 0.79 | 0.52, 0.91, 0.66 |
| $1 - \alpha_0 - \beta_0$ | 0.41 | 0.41 | 0.31 | 0.15 |
| Non-blond Acc. | 0.94 | 0.81 | 0.67 | 0.54 |
| Overall Acc. | 0.95 | 0.81 | 0.67 | 0.54 |

## A.3 ANTIGONE + JTT

Table 6: We report the (Test Average Accuracy, Test Fairness Metric) for different validation accuracy thresholds on Waterbirds dataset. We observe that Antigone + JTT (our noisy sensitive attributes) improves fairness over baseline ERM model and closes the gap with Ground-Truth + JTT (ground-truth sensitive attributes).

| Val. Thresh. | Method | DP Gap | EO Gap | Worst-group |
|---|---|---|---|---|
| [96, 96.5) | Antigone + JTT | $(95.8, 3.9) \pm (0.4, 0.4)$ | $(96.2, 10.7) \pm (0.3, 7.1)$ | $(96.3, 83.0) \pm (0.4, 1.3)$ |
|  | Ground-Truth + JTT | $(95.8, 3.9) \pm (0.4, 0.4)$ | $(96.0, 7.1) \pm (0.3, 1.4)$ | $(96.3, 83.0) \pm (0.4, 1.3)$ |
| [95.5, 96) | Antigone + JTT | $(95.4, 2.8) \pm (0.1, 0.1)$ | $(96.0, 7.5) \pm (0.2, 1.5)$ | $(96.3, 83.2) \pm (0.3, 0.6)$ |
|  | Ground-Truth + JTT | $(95.4, 2.9) \pm (0.4, 1.1)$ | $(95.6, 6.0) \pm (0.3, 2.1)$ | $(96.1, 83.5) \pm (0.5, 0.8)$ |
| [95, 95.5) | Antigone + JTT | $(94.5, 1.5) \pm (0.6, 0.6)$ | $(94.7, 4.2) \pm (0.9, 3.1)$ | $(94.7, 85.9) \pm (0.9, 1.4)$ |
|  | Ground-Truth + JTT | $(94.4, 1.7) \pm (0.7, 0.7)$ | $(94.3, 1.1) \pm (0.5, 0.6)$ | $(95.1, 86.8) \pm (0.6, 1.1)$ |
| [94.5, 95) | Antigone + JTT | $(94.2, 0.4) \pm (0.4, 0.4)$ | $(93.8, 2.0) \pm (0.5, 1.4)$ | $(94.2, 86.7) \pm (0.8, 1.8)$ |
|  | Ground-Truth + JTT | $(93.6, 0.6) \pm (0.5, 0.5)$ | $(93.8, 2.0) \pm (0.5, 1.4)$ | $(94.1, 88.2) \pm (0.6, 0.7)$ |
| [94.0, 94.5) | Antigone + JTT | $(93.0, 1.5) \pm (0.6, 0.3)$ | $(93.6, 4.8) \pm (1.2, 3.0)$ | $(93.7, 87.9) \pm (0.5, 1.4)$ |
|  | Ground-Truth + JTT | $(93.1, 1.5) \pm (0.3, 0.4)$ | $(93.2, 4.0) \pm (1.0, 2.1)$ | $(93.8, 88.1) \pm (0.7, 1.1)$ |
|  | ERM | $(97.3, 21.3) \pm (0.2, 1.1)$ | $(97.3, 35.0) \pm (0.2, 3.4)$ | $(97.3, 59.1) \pm (0.2, 3.8)$ |

Table 7: Antigone + JTT vs Ideal MC Model + JTT (Avg. Accuracy, Fairness) comparison on test data for different validation accuracy thresholds on the CelebA dataset. Lower DP and EO gaps are better. Higher WGA is better.

|  |  | DP Gap | EO Gap | WGA |
|---|---|---|---|---|
| >=94 and <95 | Antigone + JTT | (94.9, 14.7) | (94.6, 33.7) | (94.5, 61.7) |
|  | Ideal MC + JTT | (94.9, 14.7) | (94.4, 34.1) | (94.4, 58.3) |
| >=93 and <94 | Antigone + JTT | (93.7, 12.2) | (93.9, 30.3) | (93.3, 60.0) |
|  | Ideal MC + JTT | (93.7, 12.2) | (93.5, 26.3) | (93.7, 65.0) |
| >=92 and <93 | Antigone + JTT | (93.1, 12.1) | (92.4, 22.9) | (92.9, 65.6) |
|  | Ideal MC + JTT | (93.1, 12.1) | (93.0, 22.7) | (93.2, 69.4) |
| >=91 and <92 | Antigone + JTT | (91.9, 9.3) | (91.1, 13.9) | (91.1, 66.7) |
|  | Ideal MC + JTT | (91.9, 9.3) | (92.2, 19.1) | (91.8, 73.9) |
| >=90 and <91 | Antigone + JTT | (91.1, 7.9) | (91.1, 13.9) | (91.1, 66.7) |
|  | Ideal MC + JTT | (90.9, 8) | (90.4, 18.9) | (91.4, 72.2) |

Table 8: (Avg. Accuracy, Fairness) on test data for different validation accuracy thresholds on the UCI Adult dataset. Lower DP and EO gaps are better. Higher WGA is better.

| Val. Thresh. | Method | DP Gap | EO Gap | Worst-group Acc. |
|---|---|---|---|---|
| [82, 82.5) | Antigone + JTT | (81.9, 11.7) | (81.9, 0) | (81.7, 54.6) |
| | Ground-Truth + JTT | (81.8, 11.9) | (81.4, 3.3) | (81.7, 54.6) |
| [81.5, 82) | Antigone + JTT | (81.5, 9.3) | (81.5, 1.9) | (81.0, 56.0) |
| | Ground-Truth + JTT | (81.5, 9.3) | (81.1, 4.0) | (81.0, 56.0) |
| [81, 81.5) | Antigone + JTT | (80.8, 7.1) | (80.9, 0.9) | (81.0, 57.1) |
| | Ground-Truth + JTT | (80.8, 7.1) | (81.0, 2.4) | (81.0, 57.1) |
| [80.5, 81) | Antigone + JTT | (80.4, 7.2) | (80.1, 1.9) | (80.4, 58.2) |
| | Ground-Truth + JTT | (79.8, 5.1) | (80.5, 5.0) | (80.5, 58.0) |
| [80, 80.5) | Antigone + JTT | (79.7, 5.4) | (79.7, 1.4) | (80.4, 58.0) |
| | Ground-Truth + JTT | (79.5, 3.9) | (80.4, 4.2) | (80.4, 58.0) |
| | ERM | (84.8, 53.8) | (84.8, 10.6) | (84.8, 52.4) |

