# OpenReview forum: "Hyper-parameter Tuning for Fair Classification without Sensitive Attribute Access"
_ICLR.cc/2023/Conference — Submitted to ICLR 2023_

### Official Review · Reviewer_pAmd · 2022-10-21

**Confidence:** 3
**Correctness:** 4
**Technical Novelty And Significance:** 3
**Empirical Novelty And Significance:** 3
**Recommendation:** 8

**Clarity, Quality, Novelty And Reproducibility:**

The paper is clearly written and I find the proposed algorithm elegant and novel. The authors include a list of the hyper-parameters they use and provide a code for reproducibility.

**Strength And Weaknesses:**

**Strengths**

Overall, the paper is clearly written and the proposed algorithm Antigone is well-motivated. The experimental results show a significant improvement compared to previous methods. In addition, Antigone can be applied to any prior method for hyper-parameter selection, such as JTT (Liu et al., 2021). I think the method is neat and the motivation behind the EDM criterion using the mutually contaminated (MC) noise model is elegant.


**Weaknesses**

The primary weakness is that the method only applies to binary sensitive attributes. Often, sensitive attributes are non-binary, such as based on age, race, or nationality.

Second, the authors argue that their method would also work with other techniques, such as LfF (Nam et al., 2020) and CVaR DRO (Hashimoto et al., 2018a). However, they do not report any experiments to support this claim. While I don’t expect a different conclusion, it would be useful to include one or two additional methods.

Third, the hyper-parameters the authors experiment with are quite limited. They only consider three pairs of learning rate and weight decay combination and the choice of those parameters seem odd to me. A learning rate of 1e-4 is very small for a ResNet50 architecture while a weight decay of 1e-1 seems extremely large! Have the authors tried other combinations but decided not to include them because the model failed to train or did they simply stick to the choices used in JTT?

Finally, the authors experiment with two vision datasets only. Is there a reason the authors did not try additional datasets, such as Adult Income or Credit Score? It would be interesting to see how the method performs in such cases. My concern is that this line of work assumes that the error of the model is a proxy for the sensitive attribute so it would be useful to see how well this assumption holds in other settings.

**Typos**

There is a typo in Section 4: “on validation data based the EDM criterion” → “on validation data based on the EDM criterion”.


**Summary Of The Paper:**

In this paper, the authors propose an algorithm for mitigating unintended bias without requiring access to the sensitive attribute. The intuition is similar to some recent works, which assume that the errors of a model provide a noisy proxy for the sensitive attribute (e.g. minorities would have a disproportionately larger error). There have been some recent works along this direction but it has been observed that their performance was sensitive to the choice of hyper-parameters. In this paper, the authors introduce a method for hyper-parameter selection without requiring access to the sensitive attribute in both training and validation data and demonstrate experimentally (using CelebA and WaterBirds) that it performs better than previous methods.

**Summary Of The Review:**

I think the paper is novel and the experimental results show a significant improvement. The paper is clearly-written and well-motivated, particularly for such a challenging topic.

---

> ### Author Response · Authors · 2022-11-16
> **Responses and new experiments**
>
> **Q1) The primary weakness is that the method only applies to binary sensitive attributes. Often, sensitive attributes are non-binary, such as based on age, race, or nationality.**
> R1) As noted in our response to Reviewer 2, we admit that this is currently a limitation of Antigone. We note, however, that if multiple subgroups are being discriminated against, we would expect to find them over-represented in our incorrect set. From that point, we would need an additional clustering (or similar) step to disambiguate these subgroups.  We leave this as future work and have added text in the paper to reflect this limitation. We have noted this limitation in Section 2.1 and highlighted future work in the Conclusion.
>
> **Q2) Second, the authors argue that their method would also work with other techniques, such as LfF (Nam et al., 2020) and CVaR DRO (Hashimoto et al., 2018a). However, they do not report any experiments to support this claim. While I don’t expect a different conclusion, it would be useful to include one or two additional methods.**
> R2) As noted in our response to Reviewer 1, the reason we did not evaluate Antigone with LfF and CVaR DRO is because JTT was already compared with and shown to outperform LfF and CVaR DRO in the JTT paper (Liu et al. (2021)).
> However, we did implement Antigone with another fairness method GEORGE, which internally uses Group-DRO. Specifically, our comparisons with GEORGE involve replacing GEORGE’s sub-group labels on validation data with Antigone generated sub-group labels and subsequently running GEORGE’s Group-DRO implementation.
>
> **Q3) Third, the hyper-parameters the authors experiment with are quite limited. They only consider three pairs of learning rate and weight decay combination and the choice of those parameters seem odd to me. A learning rate of 1e-4 is very small for a ResNet50 architecture while a weight decay of 1e-1 seems extremely large! Have the authors tried other combinations but decided not to include them because the model failed to train or did they simply stick to the choices used in JTT?**
> R3) Yes, indeed we stuck to hyper-parameters choices reported in JTT. This was done to enable a fair comparison, but additionally, JTT alone searches over 108 different hyper-parameter settings times 50 early stopping points. This massive search space alone took us roughly ~1000 GPU hours to train and evaluate, which we did over multiple runs. Thus our primary constraint in searching over an even larger space is simply computational.
>
> **Q4) Finally, the authors experiment with two vision datasets only. Is there a reason the authors did not try additional datasets, such as Adult Income or Credit Score? It would be interesting to see how the method performs in such cases. My concern is that this line of work assumes that the error of the model is a proxy for the sensitive attribute so it would be useful to see how well this assumption holds in other settings.**
> R4) The reviewer makes a good point. We have run new experiments on the UCI Adult dataset and updated the paper with details of the experimental setup in Appendix A.1 and results in Appendix Table 8. We find that Antigone+JTT’s performance is comparable to Ground-Truth+JTT.

---

### Official Review · Reviewer_qHvW · 2022-10-24

**Confidence:** 4
**Correctness:** 3
**Technical Novelty And Significance:** 2
**Empirical Novelty And Significance:** 2
**Recommendation:** 5

**Clarity, Quality, Novelty And Reproducibility:**

I think the clarity could be considerably improved by cleaning up the notation (see comments above). The overall quality is also difficult to gauge (see comments above regarding baselines, and giving better estimates of the statistical uncertainty). The proposed method is novel, to my knowledge.

**Strength And Weaknesses:**

## Major comments

* It seems there is a clear baseline that is not evaluated here: simply use some of the (fully-labeled) training data as a held-out validation set. In that case, ground-truth attribute labels would be available. It is not clear why the paper doesn't compare to this?

* It would be helpful if the authors formally state the problem they are trying to solve (P3 simply says "we seek to train models that optimize fairness under a constraint on average accuracy").

* Similarly, it seems like the works on multicalibration/multiaccuracy are trying to solve a similar problem ("fair" classification without access to any sensitive labels); it seems particularly the fair boosting approach in (Kim et al, "Multiaccuracy: Black-box post-processing for fairness in classification") as a postprocessing method with a strong backbone model is relevant to compare to as well.

* The notation in the paper was quite confusing for me to read. There are many different sub- and superscripts, and D and X both denote (different) datasets. It also seems like \mathcal{A} is used to refer to the set of potential sensive labels (2.1) and a function class (in definition of \theta^*)? The intution of using the max-EDM model is also not clear to me; why do the authors state "we would like our noisy labeler to be biased"? It seems that *we would like to obtain accuracy pseudo-labels for the sensitive attribute* (see next comment), which is not the same thing. It is also not even clear what "bias" means in this context.

* It seems the key empirical/theoretical result for the entire paper is to show that the pseudo- attribute labels are good estimates of the true attribute labels on the validation set. The only evidence we have of this is Table 1, which strangely gives the precision (not the accuracy?) of these attribute labels. It would be useful to (1) provide the *accuracy* of these labels, (2) perhaps give some examples of data points in each (pseudo) class to see whether these match our expectation since the attribute in each dataset are easily interpretable from images, (3) perform some simulated analyses to demonstrate that under ideal conditions the pseudo-labels correctly recover the true attribute labels.

## Minor comments

* Please provide (a) the size of the data subsets in Table 1 (how many "blond men" in the validation set of CelebA?) and (b) Clopper-Pearson confidence intervals on the accuracies shown in the tables, or some other measure of statistical uncertainty of these point estimates. As is, the authors use some very fuzzy language ("JTT+Antigone is very close" and "[b]oth substantially improve upon").

* It is not clear why the method is called Antigone. This name is also not mentioned in the abstract.

* Table 3 is way too small.


## Typos etc.

There are many typos in the paper. I list a few below.

P2 "incorrectly classifier examples"

After Proposition 1: "From ... that" -> "From ... we can conclude that"

Lemma 1 "inputs data" -> input data

P7 "tune it's hyperparameters" -> tune its hyperparameters

* 3.2 why is GEORGE "a competing approach"? Competing against what? Please clarify.

**Summary Of The Paper:**

The paper presents a method for performing hyperparameter tuning without access to a sensitive attribute. Instead of obtaining ground-truth sensitive attribute labels on the validation set, the proposed method uses "pseudo" attribute labels, which are simply whether an observation is correctly predicted or not. They use these labels for performing hyperparameter tuning on JTT and George for two datasets.


**Summary Of The Review:**

Overall, the paper presents an interesting, if somewhat ad hoc seeming, approach. However, I think the exposition of the method is fairly confusing, and more importantly, I have some concerns about the empirical evaluation -- they are missing any estimates of statistical uncertainty/variability, and the empirical results are mixed and missing some clear baselines (see other comments).

---

> ### Author Response · Authors · 2022-11-16
> **Responses and new experiments**
>
> **Q1) clear baseline not evaluated: use some of the (fully-labeled) training data as a held-out validation set...ground-truth attribute labels would be available**
> R1) We compared with this baseline in the submitted paper (“JTT+True”) in Table 2. We apologize if this was not clear. We now refer to this baseline as “Ground-Truth+JTT” in Table 2,6 and 8.
>
> **Q2) It would be helpful if the authors formally state the problem they are trying to solve**
> R2) We have done so in Section 2.1 (red text).
>
> **Q3) works on multicalibration/multiaccuracy are trying to solve a similar problem…Kim et al, as a postprocessing method with a strong backbone model is relevant to compare to as well.**
> R3) Thank you for this relevant pointer. Kim at al. assume access to “a relatively small set of labeled data for auditing.” We could use Antigone to create this auditing dataset (albeit with noise); how a noisy auditing set impacts the paper’s theoretical results would merit further study. In our paper, we have used Antigone in conjunction with “in-processing” fairness methods; using Antigone with post-processing methods along with a theoretical analysis would be interesting future work. We have updated the prior work section with this discussion.
>
> **Q4) notation was confusing. There are many different sub- and superscripts, and D and X both denote (different) datasets.**
> R4) We apologize for the lack of notational clarity. We note that dataset D includes images (X), sensitive attributes (A) and target labels (Y). X was needed because the EDM metric is computed only on the images/features, so we could not use D everywhere.
> We have also updated the notation to clearly and consistently use sub/super-scripts. Specifically, previously $\gamma$ was used in both sub/super-scripts in different contexts and we now only use it as superscript. We also updated terms like $D_{1}$ to $D_{A=1}$ to clarify that we are denoting the subset of D for which A=1.
>
> **Q5) \mathcal{A} is used to refer to sensive labels (2.1) and a function class**
> R5) We apologize for the error. We have updated the draft by using \mathcal{M} for a training method in the definition of $\theta^{*}$
>
> **Q6) intution of max-EDM model not clear; why is "noisy labeler..biased"? It seems that we would like to obtain accuracy pseudo-labels for the sensitive attribute (see next comment)**
> R6) At the outset, we clarify that we are dealing with two different types of labels: known target labels, for example, hair color; and unknown sensitive attribute labels, example, gender. When we say “correct” or “incorrect” classifications, we mean in the context of the target label, i.e., hair color.
> Consider for the sake of argument a “perfectly biased” classifier on the sensitive attribute, with women as a minority group. The classifier is “perfectly biased” in the sense that it correctly classifies (misclassifies) hair color for all men (women). Then the correct (incorrect) set will contain only men (women), and we can use correct/incorrect as proxies for majority/minority groups with perfect accuracy, where accuracy is w.r.t. gender not hair color.
> In practice, we seek to intentionally amplify the bias on the validation data such that as many majority (minority) group individuals are correctly (incorrectly) classified. Since we can’t directly measure bias, we use EDM as a proxy to measure bias. We have added a new Figure 1 to provide further insight.
>
> **Q7.1) Table 1, which strangely gives the precision (not the accuracy?) of these attribute labels…useful to provide the accuracy**
> R7.1) We have updated the paper with F1-scores and accuracy numbers in Table 1.  We outperform competing methods on accuracy and F1-scores. We include sub-group F1-scores because the datasets are imbalanced, and F1-scores can be more informative in this case. The precision and recall data is moved to the appendix Table 4.
>
> **Q7.2) give examples of data points in each (pseudo) class**
> R7.2) This is a great point. We have added Figure 1 that uses the example of the CelebA dataset, where Blond Men are a minority group (only 3% of the data). The correct set for ground-truth Blond individuals (Row 1 in the figure) has only 4% Blond men, while the incorrect set  (Row 4 in the figure) has ~40% Blond men. The mean images for these sets reflect
>
> **Q7.3) simulated analyses to demonstrate that under ideal conditions the pseudo-labels correctly recover the true attribute labels.**
> R7.3) This is an excellent suggestion (raised by Reviewer 2 also). In Appendix Table 7, we compare Antigone+JTT with an Ideal MC model+JTT. We find that on DP Gap and EO Gap fairness metrics, Antigone’s results are comparable (in fact sometimes slightly better) with Ideal-MC. On WGA, we find that the Ideal MC model has a best-case WGA of 73.9% compared to Antigone’s 69.4%. Both improve significantly on the ERM model that has a WGA of 38%.

---

### Official Review · Reviewer_DQeE · 2022-10-24

**Confidence:** 4
**Correctness:** 3
**Technical Novelty And Significance:** 2
**Empirical Novelty And Significance:** 2
**Recommendation:** 5

**Clarity, Quality, Novelty And Reproducibility:**

- The reasoning of why EDM needs to be maximized is still not clear.
- The experiments are rather thin and do not fully support the claims in the paper.

**Strength And Weaknesses:**

Strengths
- Solves the realistic and challenging problem of fair training without sensitive attributes.
- The EDM maximization strategy is supported by theoretical results.
- Antigone outperforms GEORGE on real datsets.

Weaknesses
- The key assumption that there can be a bias classifier where all its correct (incorrect) predictions form the majority (minority) group seems a bit unrealistic. This means that minority group examples are extremely difficult to classify while majority group examples are much easier. If we consider the motivating examples mentioned in the Introduction like algorithmic hiring, is hiring female and male applicants that different in difficulty? I would argue that making predictions on the two sensitive groups is quite similar in difficulty, so it is not clear if the paper's assumption holds. Perhaps the assumption holds for applications where there is severe class imbalance, but in fairness problems, the gap can be more subtle than obvious. The paper does not attempt to demonstrate the key assumption empirically either. There are two real datasets CelebA and Waterbirds, and it is not clear how imbalanced their sensitive groups are.
- While the theoretical justification of EDM using the MC noise model is useful, it is not clear if the MC noise model itself really holds on real datasets. One way to verify is to add a synthetic dataset that clearly follows the MC noise model first, and then show that the Antigone results are similar on real dataset. It is often the case that real datasets have all sorts of noise that makes it difficult to understand why the proposed method is working well.
- It seems like EDM is only justified because the MC noise model is quite simplistic. Intuitively, is maximizing the Euclidean distance really all we need to do to tune the bias model? Should we not care about say the variance of the two distributions? If not, why?
- Can Antigone be extended to support more than two sensitive groups? The MC noise model seems to be limited to the binary case.
- Only experimenting on CelebA and Waterbirds seems a bit thin. What about other fairness benchmarks like AdultCensus, COMPAS, and the recent ACSIncome? More importantly, it would be informative to know how each dataset is useful in showing the advantages of Antigone. Also, it would be nice to have datasets with varying imbalances where if the imbalance is small, Antigone actually does not perform well, which is expected.


**Summary Of The Paper:**

This paper propose a fair model training method (called Antigone) without sensitive attribute access. The idea is to train a biased classifier where all its correctly-labeled examples are considered to be the majority sensitive group, and all its incorrectly-labeled examples form the minority group. The measure of success for such a biased classifier is the Euclidean distance between the means of the two groups (EDM) where a classifier with a larger EDM is best. As a justification, an mutually contaminated noise model is used where maximizing EDM results is shown to minimize DP and EO gaps. Experiments show that Antigone outperforms the competing method GEORGE.

**Summary Of The Review:**

This paper solves the challenging and important problem of fair training without sensitive attributes. There is theoretical support on why the biased classifier should maximizd EDM. However, the key assumption that training the desired biased classifier is possible is a bit questionable. In addition, the MC noise model seems quite simplistic and should be verified using the datasets. Finally, the experiments are a bit thin using only two real datasets and not fully analyzing how Antigone behaves.

---

> ### Author Response · Authors · 2022-11-16
> **Responses and new experiments**
>
> **Q1) The key assumption that minority group examples are difficult to classify while majority group examples are easier is a bit unrealistic. Consider the algorithmic hiring example in the Introduction, is hiring female and male applicants that different in difficulty? I would argue that making predictions on the two sensitive groups is quite similar in difficulty, so it is not clear if the paper's assumption holds.**
> R1) We note that results on the Adult dataset (Zemel et al. (2013)) for income prediction show discrimination on gender even though the dataset is gender balanced as a whole because high-income women are still under-represented. As noted in the paper, we do not require all majority (minority) individuals to be correctly (incorrectly) classified; just that there is a difference in their classification accuracies.
> New experiments on the Adult dataset are discussed below.
>
> **Q2) Assumption holds where there is severe class imbalance, but in fairness problems, the gap can be more subtle than obvious. Demonstrate the key assumption empirically. Clarify how imbalance the sensitive groups are on CelebA and Waterbirds**
> R2) We added Fig 2(a) to further clarify the class imbalances in CelebA and Waterbirds, with minority groups representing between 1%-3% of the training data. We additionally performed new experiments by varying the fraction of minority group individuals from {5%, 20%, 35%, 50%} in CelebA to understand the impact on the accuracy on Antigone’s pseudo-labels. The results are shown in Table 5 of the Appendix. We note as the dataset gets more balanced, the accuracy reduces (as expected) because the trained models themselves become fairer. However, even when minority group individuals (say blond men) are 35% of the total, we observe that they are still over-represented in the incorrect set vis-a-vis the correct set, i.e., still enabling noisy estimation of fairness. This can be seen from the corresponding $1-\alpha-\beta$ values (also shown in the table) which are unchanged till 20% of imbalance and still larger than zero for greater balance.
>
> **Q3) It’s not clear if the MC noise model itself holds on real datasets. Verify on a synthetic dataset that follows the MC noise model first, and then show that the Antigone results are similar on real dataset.**
> R3) This is an excellent suggestion. In Table 7, we compare Antigone+JTT with Ideal-MC+JTT. We find that on DP Gap and EO Gap fairness metrics, Antigone’s results are comparable (in fact sometimes slightly better) with those derived from the ideal MC model. On WGA, the most challenging fairness metric to optimize for, we find that the Ideal MC model has a best-case WGA of 73.9% compared to Antigone’s 69.4%.
> Recall that with Ground-Truth validation data, the WGA is 78.2% while with standard ERM it is only 38%. As such, we find, as expected, that Ground-Truth>Ideal-MC>Antigone>ERM.
>
> **Q4) It seems like EDM is only justified because the MC noise model is quite simplistic. Intuitively, is maximizing the Euclidean distance really all we need to do to tune the bias model? Should we not care about say the variance of the two distributions? If not, why?**
> R4) Indeed, the EDM metric is theoretically justified in the asymptotic regime wherein the fairness metrics on noisy sensitive labels converge to (1-\alpha-\beta)*(fairness on true sensitive labels). In practice, with a finite validation set, the variance will matter but can be challenging to estimate and account for. See for example recent results that analytically derive the variance for fairness metrics (Mozannar et al., 2020). We leave the possibility of deriving similar bounds for fairness under the MC noise model.
> We do note that it is common practice to use the mean accuracy/fairness measured on the validation set for hyper-parameter tuning. Further, although more sophisticated measures of distance between distributions have been proposed in literature (e.g. https://arxiv.org/pdf/2002.02923.pdf) and could be potentially used, these metrics are computationally prohibitive. As such, to some extent we view the simplicity of the mean distance metric as a feature of our method.
>
> **Q5) Can Antigone support more than two sensitive groups?**
> R5) This is currently a limitation of Antigone. We note, however, that if multiple subgroups are being discriminated against, we would expect to find them over-represented in our incorrect set. From that point, we would need an additional clustering (or similar) step to disambiguate these subgroups.  We leave this as future work and have added text in the paper to reflect this limitation.
>
> **Q6) Add experiments on other fairness benchmarks**
> R6) We added new experiments on the Adult dataset and updated the paper with details of the experimental setup in Appendix A.1 and results in Appendix Table 8. Standard ERM models have large DP and EO gaps that can be reduced using Antigone+JTT. We find that Antigone+JTT’s performance is comparable to Ground-Truth+JTT.

---

### Official Review · Reviewer_iLMB · 2022-10-25

**Confidence:** 4
**Correctness:** 3
**Technical Novelty And Significance:** 3
**Empirical Novelty And Significance:** 2
**Recommendation:** 5

**Clarity, Quality, Novelty And Reproducibility:**

The paper is clearly written, but several clarifications are needed as discussed in the previous section.

**Strength And Weaknesses:**

[Strength]

S1: The paper tackles the practical problem of accessing sensitive attributes in the validation set.

S2: The paper utilizes an idea from fair training under noisy sensitive attributes, and such inspiration gives an interesting approach.


[Weakness]

W1: The significance of the proposed algorithm seems a bit limited, as it only targets fairness techniques that do not access sensitive attributes in training data. Since many more algorithms have been proposed for fair training with sensitive attributes, it would be helpful if the paper can clarify 1) the significance of this work and 2) how the proposed algorithm can be extended in other scenarios.

W2: Several design choices in the proposed framework Antigone are not fully justified. For example, why using the previous idea of handling noisy group attributes is the most appropriate way to solve the target problem? Why the current approach is better than other approaches like training a weak-labeler on groups?

W3: More importantly, the experimental results are insufficient.
- The baselines are not enough. One of the key advantages the paper argues is that Antigone can be used for any algorithm of not accessing sensitive attributes on training data. However, the paper only shows its effectiveness on JTT. Thus, it is hard to believe that Antigone will effective in other algorithms like LfF and DRO.
- Also, all experimental results are reported without error range, which makes the observations less reliable. For example, in several rows of Table 1, the improvements in Antigone compared to GEORGE are a bit marginal, so it is questionable how it changes after multiple runs. It would be much better if the paper shows the results with error ranges to clarify the performance gain.

Minor Typo: First paragraph in Section 2.1) target labels (X) => target labels (Y)


**Summary Of The Paper:**

The paper focuses on the problem of accessing sensitive group attributes in the validation set for hyperparameter tuning. To solve this issue, the paper proposes Antigone, a hyper-parameter tuning framework for fairness techniques that do not access sensitive group attributes in training data. The proposed framework is inspired by a prior work (Lamy et al., 2019), which has been proposed for fair training under noisy sensitive attributes. This algorithm is evaluated on two datasets, including CelebA and Waterbirds, with some baselines.

**Summary Of The Review:**

Although the paper tries to solve a practical problem, there seems some room for improvement in 1) clarification on the importance of the work, 2) justification of the design choices, and 3) enhancement of the experimental results. Overall, I think that this paper is on the borderline.

---

> ### Author Response · Authors · 2022-11-16
> **Responses and new experiments**
>
> **Q1) Clarify the significance of this work.**
> R1) While prior work often assumes access to sensitive attributes, real-world settings in which sensitive attributes may not be available have been highlighted by a growing body of literature (Veale & Binns, 2017; Holstein et al., 2019). This is for multiple reasons, as we note in the paper. “For example, data subjects may abstain from providing sensitive information to eschew potential discrimination in future (Markos et al., 2017). In other settings, the attributes on which the model discriminates might not even be known (Citron & Pasquale, 2014; Pasquale, 2015). For instance, in algorithmic hiring decisions, K ̈ochling & Wehner (2020) highlight that bias and discrimination are often recognized only after making real world decisions on applicants due to unknown attributes on which the model discriminates during training. Consequently, a large American e-commerce company had to cease using algorithmic tools for hiring purposes as it was unintentionally discriminating female applicants (Dastin, 2018).”
> Our work (and prior work including JTT and GEORGE) seeks to mitigate harms in these real-world settings which, as the literature above suggests, do happen commonly.
>
> **Q2) How the proposed algorithm can be extended in other scenarios?**
> R2) We note that prior work like JTT also addresses the problem of spurious correlations (Sagawa et al., 2020; Wang & Culotta, 2021). As such Antigone can also be used in conjunction with JTT for addressing spurious correlations; here the “incorrect” set will have an over-representation of data that is misclassified due to these spurious correlations. We have added a note in the introduction to reflect this.
>
> **Q3) Several design choices in the proposed framework Antigone are not fully justified. For example, why using the previous idea of handling noisy group attributes is the most appropriate way to solve the target problem? Why the current approach is better than other approaches like training a weak-labeler on groups?**
> R3) We note that sensitive group information is unavailable on training/validation data in our problem setting. Therefore, we cannot train a supervised classifier to predict group labels because ground-truth group labels are missing. Hence, we have to treat the problem of recovering group labels as an unsupervised learning problem. For example, our competing work, GEORGE, uses k-means clustering, a standard unsupervised learning method for this task.
> Anitgone improves upon GEORGE by recognizing that a supervised classifier trained on target labels (note that unlike group labels, target labels are available during training) is likely to make more mistakes on minority versus majority groups. Thus correctly/incorrectly classified examples can serve as proxies for majority/minority groups. We use the MC noise model to derive a metric that helps us pick a classifier that yields more accurate group labels compared to GEORGE.
>
> **Q4) The baselines are not enough…paper only shows its effectiveness on JTT. Thus, it is hard to believe that Antigone will effective in other algorithms like LfF and DRO.**
> R4) We highlight that we have evaluated Antigone with both JTT and with Group-DRO in our evaluations of Antigone+GEORGE. Specifically, our comparisons with GEORGE involve replacing GEORGE’s sub-group labels on validation data with Antigone generated sub-group labels and subsequently running GEORGE’s Group-DRO implementation. The reason we did not evaluate Antigone with LfF is because JTT was already compared with and shown to outperform LfF in the JTT paper (Liu et al. (2021)).
>
> **Q5) Also, all experimental results are reported without error range, which makes the observations less reliable. For example, in several rows of Table 1, the improvements in Antigone compared to GEORGE are a bit marginal, so it is questionable how it changes after multiple runs.**
> R5) We updated the tables in the paper with error ranges from multiple runs. Table 1 (updated to show F1-score and accuracy instead of precision as suggested by Reviewer 3) shows that the errors (over five runs) are small and Antigone consistently outperforms GEORGE on all but one datapoint for F1 score, and on overall accuracy. Table 3 shows our overall comparisons with baseline GEORGE vs. Antigone+GEORGE. Antigone+GEORGE has higher worst-group accuracy (WGA) on both the CelebA and Waterbirds dataset. In interpreting the errors, we should note that Antigone+GEORGE’s WGA was equal to or better than GEORGE’s in each one of our multiple runs.
> In Table 2, we also show errors for Antigone+JTT over four runs. Our primary limitation in obtaining more data was computational constraints. Each run takes ~1000 GPU hours for JTT and ~10 GPU for GEORGE. We have additional runs in progress and will update the paper with the additional runs by the Friday deadline.

---

### Decision · Program_Chairs · 2023-01-20

**Decision:**

Reject

**Justification For Why Not Higher Score:**

Not validating the assumptions or motivating why the method should work.

**Justification For Why Not Lower Score:**

N/A

**Metareview: Summary, Strengths And Weaknesses:**

The paper presents a new and effective way to tune the hyperparameters of fairness procedures without requiring access to sensitive attributes.

The reviewers appreciated the proposed method that provides less dependence on sensitive attributes which may not be available in practice or expensive to obtain. The reviewers also appreciated the promising empirical results.

However, several reviewers pointed out that the paper does not validate or motivate the assumptions and reasoning behind why the method should work. Such would be required to complete the paper, and as such, was the key reason for rejection.